# Prognostic variables and 4-year survival outcomes in CD20 Positive AIDS-Related Lymphoma in the Anti-retroviral treatment era: A Retrospective Review from a Single Centre in KwaZulu-Natal, South Africa

**Nadine Rapiti**[1]*, **Nada Abdelatif**[2], **Mahomed-Yunus S. Moosa**[3]

**1** Department of Haematology, NHLS/University of KwaZulu Natal/King Edward VIII Hospital, Durban, South Africa, **2** Biostatistics Research Unit, South African Medical Research Council, Cape Town, South Africa, **3** Department of Infectious Diseases, University of KwaZulu-Natal, Durban, South Africa

* rapitin@ukzn.ac.za

## Abstract

### Objective

To describe 4-year survival outcomes and assess the value of established and additional relevant variables to predict complete response (CR), four-year progression-free survival (PFS) and overall survival (OS) of CD20 positive AIDS-Related Lymphoma (ARL) treated with standard combination chemotherapy.

### Method

We performed a retrospective review of patients diagnosed with CD20 positive ARL between 2006 and 2016. All patients over 12 years of age who received at least one cycle of combination chemotherapy with curative intent were included in the analysis. Variables assessed included the International Prognostic Index (IPI), age-adjusted-IPI, age, gender, B symptoms, extent of disease, functional performance status, CD4 cell count, viral load, concurrent ART with chemotherapy, rituximab inclusion, and number of chemotherapy cycles used. Kaplan-Meier survival curves for OS and PFS at 4 years were compared for IPI and aaIPI using the log-rank test. A Cox proportional hazards model was used to investigate the effects of prognostic variables for patients achieving OS and PFS at 4 years and logistic regression for patients achieving CR.

### Results

A total of 102 patients were included in the analysis. At year four of follow-up, the OS was 50% (n = 51) and PFS was 43% (n = 44). Attaining a CR and male gender were significantly associated with improved 4-year OS (p<0.001 and p = 0.028 respectively) and PFS (p<0.001 and 0.048 respectively). A viral load of < 50 copies/ml was associated with a higher complete response rate (aOR 6.10 [95% CI 1.15, 24.04], p = 0.01). Six or more cycles of

**Data Availability Statement:** All relevant data are within the article and its Supporting Information files.

**Funding:** The study was funded by Roche Pharmaceuticals; funding to NR. The fund was managed by the University of KwaZulu-Natal. The funders had no role in study design, data collection and analysis, decision to publish, or preparation of the manuscript.

**Competing interests:** The authors have declared that no competing interests exist.

**Abbreviations:** CR, Complete response; PR, Partial response; PFS, Progression free survival; OS, Overall survival; ARL, AIDS related lymphoma; NHL, Non-Hodgkin lymphoma; HIV, Human immunodeficiency virus; DLBCL, Diffuse large B cell lymphoma; BL, Burkitt lymphoma; PL, Plasmablastic lymphoma; PEL, Primary effusion lymphoma; CNS, Central nervous system; PCNSL, Primary CNS lymphoma; IPI, International prognostic index; aaIPI, Age adjusted IPI; aOR, Adjusted odds ratio; aHR, Adjusted hazard ratio; DLBCL, Diffuse large B cell lymphoma; ART, Antiretroviral therapy; HAL, HIV associated lymphoma; LDH, Lactate dehydrogenase; CHOP, Cyclophosphamide, doxorubicin, vincristine, prednisone; ECOG, Eastern Cooperative Oncology Group; CNS, Central nervous system; MINE, mesna, ifosfamide, novantrone, etoposide; VL, Viral load.

chemotherapy was superior to fewer cycles for both PFS (aHR 0.17 [95% CI 0.10, 0.29]) and OS (aHR 0.12 [95% CI 0.07, 0.22]) with p-value < 0.001 for both PFS and OS. The Kaplan-Meier survival estimates demonstrated the prognostic utility of the IPI and aaIP for OS (p = 0.002 and 0.030 respectively) and the IPI for PFS (p = 0.002).

## Conclusion

This study is a first from a high prevalence HIV area in KwaZulu-Natal, South Africa, and confirms the utility of the internationally accepted prognostic scoring systems in predicting survival in CD20 positive ARL in the local population.

## Introduction

Lymphomas that occur more frequently in people living with human immunodeficiency virus (HIV) are referred to as HIV-associated lymphomas (HAL). Three malignancies are considered AIDS-defining: high-grade non-Hodgkin lymphoma (NHL), Kaposi sarcoma and invasive cancer of the cervix. The AIDS-defining high-grade NHL are collectively called AIDS-related lymphomas (ARL) and are aggressive with diverse histologic characteristics [1]. These ARL include diffuse large B cell lymphoma (DLBCL), Burkitt lymphoma (BL), plasmablastic lymphoma (PBL) and less commonly, primary CNS lymphoma (PCNSL) and primary effusion lymphoma (PEL) [2,3]. DLBCL and BL usually express CD20, and the other ARL are generally CD20 negative. The introduction of antiretroviral therapy (ART) in 1996 not only reduced the incidence of ARL but also improved outcomes [4–6]. Despite the widespread use of ART, ARL remains a common malignancy and a primary cause of AIDS-related malignant deaths [7–9]. South Africa is home to just over 20% of the global population living with HIV [10]. KwaZulu-Natal is a province in South Africa that has an HIV prevalence of 18% [11]. Ascertaining reliable and valid prognostic tools for rational utilization of scarce resources for the management of ARL is critical, particularly for these resource-limited settings.

Outcomes in ARL, both CD 20 positive and CD20 negative, are associated with characteristics of the host, malignancy, and HIV disease. The former two variables are incorporated into the well-established international prognostic index (IPI), which consists of the patient's age, performance status, lactate dehydrogenase (LDH) level, stage of lymphoma and the number of extranodal sites involved [12]. This index for aggressive lymphoma was initially derived from an HIV-negative cohort treated with combination chemotherapy [12]. This scoring system was subsequently validated for ARL treated with standard cyclophosphamide, doxorubicin, vincristine, prednisone (CHOP) chemotherapy as well as chemotherapy supplemented with CD20 monoclonal antibodies (rituximab) [13–15]. Another scoring index, the age-adjusted international prognostic index (aaIPI), is a modified IPI score utilizing three variables of the IPI, viz. performance status, LDH level and stage of disease, strongly correlates with outcomes of ARL [12]. This score was validated in a pooled analysis of 1546 patients from 19 prospective trials conducted in the United States and Europe [16]. This study population included predominantly male patients, with a good median CD4 count, and differs from the stronger female representation and poorer CD4 count in ARL described in two South African studies as well as East Africa [17–19].

Unlike the IPI and aaIPI, the HIV-specific prognostic variables for ARL are less well-defined. Studies have variably analysed the prognostic significance of the viral load, CD4 count and history of prior AIDS-defining illness, showing poorer outcome with the latter two

variables [20,21]. However, the significance of these variables has evolved over time with the use of ART, with less prognostic significance attributed to the CD4 count [22]. In this post-ART era, it is unclear if a specific ARL prognostic score would better determine prognosis in ARL than the IPI or aaIPI, and which variables would need to be included in such a score [23].

The therapeutic advances in ARL have resulted in outcomes comparable to HIV-negative patients, spurring the need to optimise current prognostic tools to improve management [13,20,24]. This is especially pertinent in countries with a high HIV prevalence where a significant portion of the health care budget is utilised in the management of ARL. The IPI and aaIPI prognostic scoring systems have not been validated in the local cohort or in the three other South African studies on ARL [17,18,25]. In KwaZulu-Natal, the significance of other lymphoma-related, or HIV-associated variables, including the CD4 count, concurrent ART and viral load in CD20 positive ARL is also unknown. In these settings, accurate prognostic tools for an African cohort will guide the implementation of risk-adapted, cost-effective treatment strategies.

## Methods

This was a retrospective review of patients with histologically confirmed CD20 positive ARL managed at King Edward Vlll Hospital between January 2006 and December 2016. To be included, patients had to be over the age of 12 years, have received at least one cycle of CHOP chemotherapy, with or without rituximab (R), with curative intent, and have had at least four years of follow-up. Patients with Burkitt lymphoma were not included in this study, as these patients are treated with a more intensive chemotherapy protocol. Patients with primary central nervous system (CNS) lymphoma were also excluded from this analysis. Anonymized data from the patients' haematology charts was captured onto an Excel electronic spreadsheet. Laboratory data was further obtained from the National Health Laboratory Service laboratory information system. Data was exported into Stata 16 programme for analysis. The study protocol was approved by the Biomedical Research Ethics Council of the University of KwaZulu-Natal (BE043/17), and the study complied with the principles of the Declaration of Helsinki.

All patients in the study cohort were managed according to standard lymphoma treatment guidelines [26,27]. Between 2006 and 2013, CHOP was the main chemotherapy regimen. During 2013 local guidelines added rituximab to CHOP chemotherapy. CHOP and RCHOP were given at standard doses, except for prednisone, which was given at a dose of 60mg daily, orally for 8 days rather than 100mg daily for 5 days, according to the centre's practice. Here we report on the effect of established and individual prognostic variables on CR, 4-year OS and 4-year PFS. Patient-related variables considered for their prognostic value included age, gender, and Eastern Cooperative Oncology Group (ECOG) functional status. Tumor-related variables considered included B symptoms, Ann Arbor stage, use of rituximab, number of chemotherapy cycles, radiotherapy and CNS involvement. HIV-specific variables included baseline CD4 cell count, viral load, ART status, and timing of ART relative to initiation of chemotherapy. The cut-off for the age of 40 and CD4 of 100 cells/μL was chosen to allow for comparison with existing data, which shows prognostic significance to these variables at these limits [16,21,25,28]. The IPI score was categorized as low risk (0–2), and high risk (3–5). An aaIPI of 0–1 was considered low risk and a score of 2–3 as high risk.

CNS involvement was determined by imaging or cerebrospinal fluid examination using cytocentrifuge or flowcytometry. Response to therapy was assessed at mid-cycle and at completion of chemotherapy. CR was defined by the absence of disease assessed by clinical, radiological and laboratory measures. All records were available, except for viral load, where only 65 patients had information.

## Statistical analysis

Kaplan-Meier survival curves for OS and PFS at 4 years were compared for IPI and aaIPI (low versus high risk) using the log-rank test. Cox proportional hazards model was used to investigate the effects of prognostic variables on patients achieving OS and PFS during the study period. Logistic regression was used to determine the effects for the different prognostic variables in patients who achieved CR. Unadjusted and adjusted hazards and odds ratios are reported. All variables and outcome measures were evaluated on an intention-to-treat basis. All time to event analyses were measured from presentation to event or last recorded follow-up. P-values of <0.05 were considered statistically significant. Stata 16 was used for all data analyses [29].

## Results

During the study time period, 157 ARL were managed at this hospital. There were 102 patients with CD20 positive ARL included in this analysis (histological subtypes were 70 DLBCL, 31 high-grade B cell lymphoma, and 1 high-grade B cell lymphoma not otherwise specified). 13 patients were excluded due to missing data (either histology or HIV results), 9 patients for receiving an alternative chemotherapy regimen, 5 patients who did not receive chemotherapy and 2 patients with primary effusion lymphoma. The data for 26 plasmablastic patients has been published [30]. The chemotherapy regimens and 4-year survival outcomes are shown in **Fig 1**. From the 102 patients, 50% (n = 51) were alive at 4 years, with 43% (n = 44) showing no disease progression, 12% (n = 12) demising and 38% (n = 39) were lost to follow up during the 4 years. 64% (25/39) of the patients defaulting treatment were lost to follow-up within a year and none of these patients were in a CR at the time. There were 5 patients defaulting treatment between 12–24 months, and of the nine patients lost to follow-up after 2 years, 4 were treated with CHOP (2 with a partial response [PR] who chose not to continue further therapy, and 2 in CR) and 5 were treated with R-CHOP (all in CR). There were 3 patients who relapsed post first-line chemotherapy, with 2 receiving salvage($_s$) therapy and the third patient was lost to follow-up immediately after documented relapse. For 21 patients receiving salvage therapy with mesna, ifosfamide, novantrone, etoposide (MINE), and/or high dose methotrexate and/ or radiotherapy, 9 achieved a CR$_s$, 9 were lost to follow-up and 3 demised.

For the patient and HIV prognostic variables shown in **Table** 1, the HIV viral load was the only variable that significantly affected the achievement of CR. Patients having undetectable viral loads of <50 copies/ml had a greater likelihood of CR (adjusted OR 6.10 [95% CI 1.55, 24.04]). Attaining a CR was significantly associated with improved PFS (HR = 0.08 [CI: 0.04, 0.16]; p-value <0.001) and OS (HR = 0.11 [CI: 0.06, 0.22)]; p-value <0.001). At lymphoma diagnosis, 45% (n = 46) of the patients were already on ART, 39% (n = 40) initiated ART concurrently with chemotherapy and 16% (n = 16) remained ART naïve during chemotherapy. Of the 86 patients who received concurrent ART with chemotherapy, 60% (n = 50) achieved a CR and 25% (n = 4) of the ART naïve group achieved a CR (p = 0.994). Of the 16 ART naïve patients, 12 received CHOP. Four of these 12 patients attained a CR and two a PR, with five of these patients having an OS exceeding five years and the sixth patient defaulted therapy after 10 months. Of the remaining six patients on CHOP, one had disease progression, two demised after three months, and three defaulted follow-up, with OS ranging from 2 weeks to six months. Four ART naïve patients received R-CHOP; one had disease progression and died 25 months later, and the three remaining patients were lost to follow-up during chemotherapy (none in CR). Using the chi-squared test, the 4-year OS of 59% (46/86) for concurrent ART with chemotherapy and 31% (5/16) for those who remained ART naïve during chemotherapy, was not significantly associated with improved OS(p = 0.111).

**Total cohort n=102**

Demise n=9
LTFU n=11
PR n=7
DP n=19

Demised n=12
LTFU n=30

Demise n=12
LTFU n=30
DP n=7

CR$_1$ n=56 (55%)

2-year OS n=60 (59%)

2-year PFS n=53 (52%)

Relapse n=3

LTFU n=9

LTFU n=11

LTFU n=9

CR$_1$ at 4 years n=42 (41%)

4-year OS n=51 (50%)
CHOP n=25
R-CHOP n=26

4-year PFS n= 44 (43%)
CHOP n=20
R-CHOP n=24

Died n=3

Salvage therapy n=21 (21%)

Total CR (CR$_1$ + CR$_s$) at 4 years n=51 (50%)

LTFU n=9

**Fig 1. Treatment and outcomes for CD20 positive ARL.**

**Table 1. Impact of patient and HIV variables on CR.**

| | | | CR (n = 56) | |
| --- | --- | --- | --- | --- |
| **Predictor (%)** | | **Unadjusted OR (95% CI)** | **Adjusted OR (95% CI)** | **p-value** |
| **Age in years** | <40 (55%)<br>≥40 (45%) | ref<br>1.32 (0.60, 2.91) | ref<br>0.88 (0.26, 3.04) | ref<br>0.841 |
| | Female (54%) | ref | ref | ref |
| **Gender** | Male (46%) | 1.03 (0.47, 2.26) | 1.70 (0.51, 5.62) | 0.386 |
| **B symptoms** | Absent (70%) | ref | ref | ref |
| | Present (30%) | 0.83 (0.35, 1.93) | 0.84 (0.22, 3.24) | 0.798 |
| **ART timing** | Pre-diagnosis (53%) | ref | ref | ref |
| | Concurrent with chemotherapy (47%) | 0.79 (0.33, 1.89) | 1.01 (0.25, 4.11) | 0.994 |
| **CD4 count in cells/μL** | < 100 (25%) | ref | ref | ref |
| | ≥ 100 (75%) | 1.60 (0.66, 3.93) | 1.15 (0.18, 7.52) | 0.885 |
| **VL** | Undetectable or <50 copies/ml (49%) | 8.21 (2.68, 25.17) | 6.10 (1.55, 24.04) | 0.010 |
| | ≥50 copies/ml (51%) | ref | ref | ref |

Abbreviations: CR = complete response, OR = odds ratio, ART = antiretroviral therapy, VL = viral load.

**Table 2. Impact of patient and HIV variables on outcome for CD20 positive ARL.**

| Predictor (%) | PFS | | | OS | | |
|---|---|---|---|---|---|---|
| | Unadjusted HR (95% CI) | Adjusted HR (95% CI) | p- value | Unadjusted HR (95% CI) | Adjusted HR (95% CI) | p-value |
| **Age in years** <40 (55%) ≥40 (45%) | ref | ref | ref | ref | ref | ref |
| | 0.91 (0.54, 1.52) | 1.02 (0.47, 2.22) | 0.952 | 0.93 (0.53, 1.61) | 0.95 (0.43, 2.10) | 0.906 |
| **Gender** Female (54%) Male (46%) | ref | ref | ref | ref | ref | ref |
| | 0.77 (0.46, 1.29) | 0.46 (0.21, 0.99) | 0.048 | 0.78 (0.45, 1.37) | 0.41 (0.19, 0.91) | 0.028 |
| **B symptoms** Absent (70%) Present (30%) | ref | ref | ref | ref | ref | ref |
| | 1.04 (0.59, 1.81) | 0.95 (0.41, 2.21) | 0.901 | 1.18 (0.66, 2.12) | 1.02 (0.43, 2.41) | 0.959 |
| **ART** Pre-diagnosis (53%) Concurrent with chemotherapy (47%) | ref | ref | ref | ref | ref | ref |
| | 1.01 (0.56, 1.82) | 0.84 (0.35, 2.04) | 0.702 | 0.96 (0.51, 1.78) | 0.85 (0.34, 2.11) | 0.719 |
| **CD4 count in cells/µL** < 100 (25%) ≥ 100 (75%) | ref | ref | ref | ref | ref | ref |
| | 0.81 (0.45, 1.44) | 1.28 (0.38, 4.35) | 0.694 | 0.93 (0.50, 1.74) | 1.18 (0.35, 4.06) | 0.787 |
| **VL** Undetectable or <50 copies/ml (49%) ≥50 copies/ml (51%) | 0.45 (0.23, 0.88) | 0.45 (0.19, 1.07) | 0.070 | 0.41 (0.21, 0.82) | 0.43 (0.18, 1.05) | 0.063 |
| | ref | ref | ref | ref | ref | ref |

Abbreviations: ARL = AIDS related lymphoma, HR = hazard ratio, CR = complete response, PFS = progression free survival, OS = overall survival, ART = antiretroviral therapy, VL = viral load.

For the 56 patients that attained a CR at the end of first line chemotherapy, two patients received 3 cycles, one had 4 cycles and another 5 cycles of chemotherapy (all four patients having refused additional chemotherapy). Of the 9 patients who received radiotherapy, 83% on CHOP (5/6) and 67% on R-CHOP (2/3) achieved a 4-year OS respectively. The impact of patient and HIV variables on PFS and OS are detailed in **Table 2** and lymphoma and treatment-related variables in **Table 3**.

Of the total cohort, 9% (n = 9) received radiotherapy; 4 patients with bulky disease, 1 patient with a tonsillar relapse (who received salvage chemotherapy and radiotherapy) and progressive disease treated with salvage chemotherapy in the remaining four patients. A third of the study patients who received radiotherapy (3/9) had a good response with the sole addition of curative dose radiotherapy to 1st line chemotherapy, converting partial responses to CR in all three patients (all 3 had bulky disease). There was a 78% (7/9) 4-year OS for patients receiving radiotherapy compared with 47% (44/93) for patients who did not receive radiotherapy (p = 0.071).

There was no significant association between individual prognostic variables of the aaIPI and OS or PFS (**Table 4**). However, the IPI and aaIPI significantly correlated with OS (**Figs 2 and 3** respectively) and the IPI also significantly correlated with PFS and not OS (**Fig 2**).

## Discussion

Better HIV control in ARL has allowed for chemotherapy intensification with monoclonal antibodies similar to HIV-negative patients with lymphoma, and this is the first study from

**Table 3. Lymphoma and treatment related variables in CD20 positive ARL.**

| Predictor (%) | PFS | | | OS | | |
|---|---|---|---|---|---|---|
| | Unadjusted HR (95% CI) | Adjusted HR (95% CI) | p- value | UnadjustedHR (95% CI) | Adjusted HR (95% CI) | p-value |
| **CNS involvement by lymphoma**<br>Absent (93%)<br>Present (7%) | ref<br>1.20<br>(0.48, 3.00) | ref<br>1.04<br>(0.35, 3.10) | ref<br>0.951 | ref<br>1.04 (0.38) | ref<br>1.58<br>(0.49, 5.04) | ref<br>0.444 |
| **Rituximab use**<br>No (49%)<br>Yes (51%) | ref<br>0.81<br>(0.48, 1.36) | ref<br>0.76<br>(0.45, 1.27) | ref<br>0.289 | ref<br>1.04<br>(0.60, 1.80) | ref<br>0.89<br>(0.51, 1.55) | ref<br>0.675 |
| **Radiotherapy use**<br>No (91%)<br>Yes (9%) | ref<br>0.86<br>(0.34, 2.16) | ref<br>0.70<br>(0.23, 2.09) | ref<br>0.519 | ref<br>0.33<br>(0.08, 1.37) | ref<br>0.23<br>(0.05, 1.14) | ref<br>0.071 |
| **Number of chemotherapy cycles**<br>< 6 (30%)<br>6–8 (70%) | ref<br>0.16<br>(0.10, 0.29) | ref<br>0.17<br>(0.10, 0.29) | ref<br><0.001 | ref<br>0.13<br>(0.07, 0.23) | ref<br>0.12<br>(0.07, 0.22) | ref<br><0.001 |

Abbreviations: ARL = AIDS related lymphoma, HR = hazard ratio, ARL = AIDS related lymphoma, CR = complete response, PFS = Progression free survival, OS = overall survival, CNS = central nervous system.

South Africa describing the prognostic significance of rituximab use in ARL. This study 4-year OS of 50% is within the reported 5-year survival of 46–55% for ARL [20,24] and comparable to the reported 46% 5-year survival in another South African study [18]. This improved 4-year survival in the ART era, compared with the 11-month median OS described in Gauteng, South Africa with 85% of the cohort not on ART [17], highlights the need for an accurate prognostic system to guide therapy in ARL.

The CR for the study cohort of 55% is at the lower spectrum of reported CR; 65% described by Miralles et al in the multicentre Spanish study [20] and 58–77% in American and European studies [31,32]. However this CR correlated with 4-year OS and PFS in our cohort. This finding is in accordance with the positive predictive value of the CR for OS found in a retrospective British cohort [13].

**Table 4. aaIPI prognostic variables on outcome of CD20 positive ARL.**

| Predictor (%) | PFS | | | OS | | |
|---|---|---|---|---|---|---|
| | Unadjusted HR (95% CI) | Adjusted HR (95% CI) | p- value | Unadjusted HR (95% CI) | Adjusted HR (95% CI) | p-value |
| **Ann Arbor stage**<br>Early—I/II (55%)<br>Late—III/IV (45%) | ref<br>1.92<br>(1.14, 3.23) | ref<br>1.93<br>(0.95, 3.90) | ref<br>0.067 | ref<br>2.05<br>(1.17, 3.59) | ref<br>2.04<br>(0.96, 4.37) | ref<br>0.065 |
| **ECOG PS**<br>< 2 (32%)<br>≥ 2 (68%) | ref<br>1.66<br>(0.91, 3.04) | Ref<br>1.05<br>(0.47, 2.35) | ref<br>0.899 | ref<br>1.70<br>(0.89, 3.25) | ref<br>1.03 (0.43, 2.46) | ref<br>0.941 |
| **LDH increased**<br>No (4%)<br>Yes (96%) | ref<br>0.71<br>(0.22, 2.26) | Ref<br>0.56<br>(0.17, 1.85) | ref<br>0.340 | ref<br>1.01<br>(0.24, 4.14) | ref<br>0.77<br>(0.18, 3.27) | ref<br>0.727 |

Abbreviations: ARL = AIDS related lymphoma, aaIPI = age adjusted International Prognostic Index, CR = complete response, PFS = progression free survival, OS = overall survival, HR = hazard ratio, ECOG PS = Eastern Cooperative Oncology group performance status, LDH = lactate dehydrogenase.

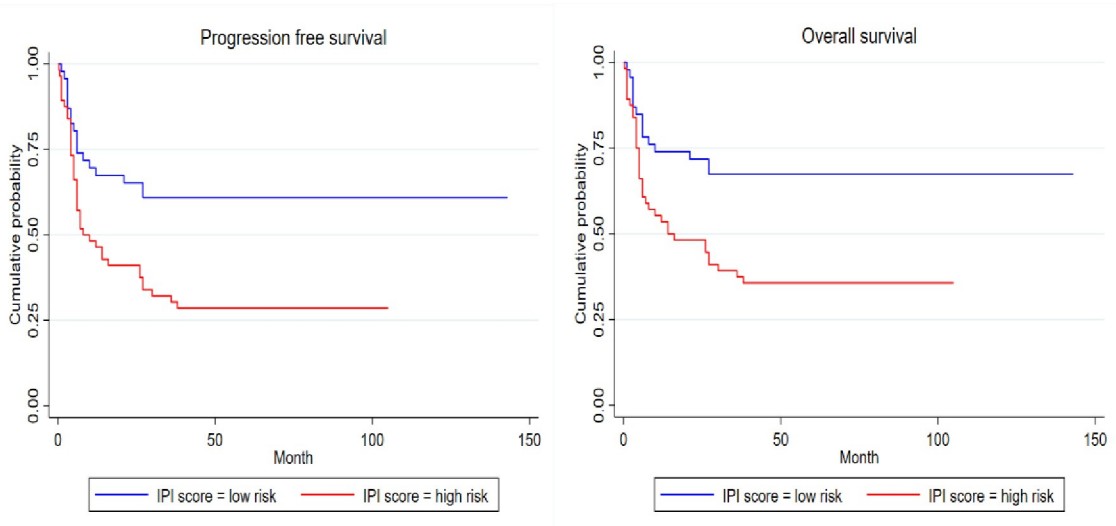

**Fig 2. Kaplan Meier survival estimates: IPI for for PFS, p-value = 0.002 (left) and IPI for OS, p-value = 0.002 (right).**

The only patient-related variable that correlated with survival, was gender, with males having both improved PFS and OS. This prognostic significance of gender has been confirmed by other studies in Africa, including South Africa [18,33]. These findings, however, contrast with the poorer outcome shown for male patients in Zimbabwe, and lack of significance for gender in studies outside of Africa [28]. Whether gender remains a unique prognostic variable for an African cohort in the ART era warrants further study.

There was no improvement in 4-year OS in patients who received ART with chemotherapy compared with those who remained ART naïve during chemotherapy (p = 0.111). This result contrasts with the Lim et al [21] analysis which showed that median survival increased significantly from 8.3 months in ART naïve patients to 43.2 months for those on ART. Our study findings also contrast with an African study from Uganda which found similar survival

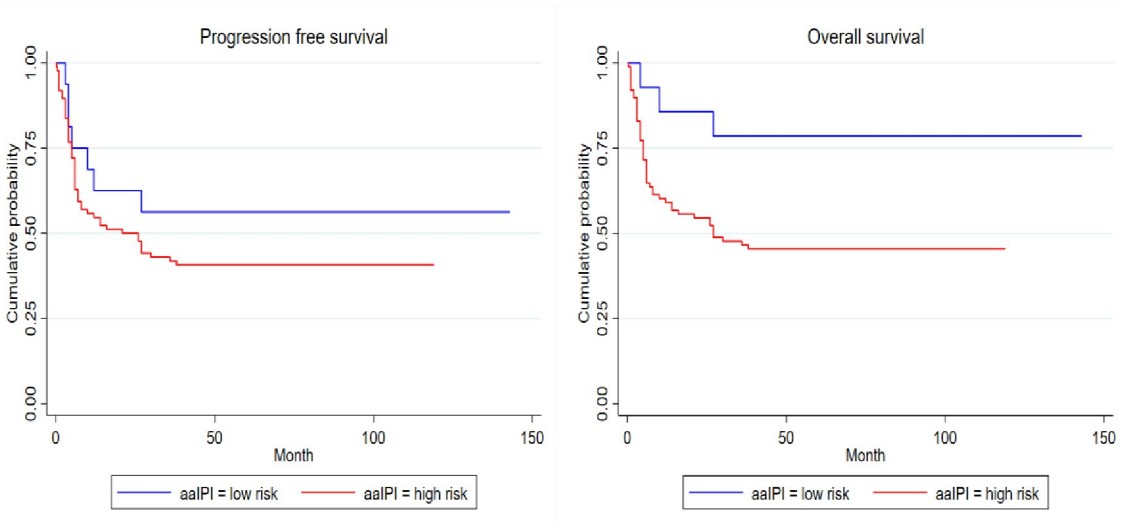

**Fig 3. Kapan Meier survival estimates: aaIPI for PFS, p-value = 0.266 (left) and aaIPI for OS, p-value = 0.030 (right).**

outcomes in ARL treated with concurrent ART and chemotherapy compared with HIV-negative lymphoma patients, but an almost 9-fold increase in mortality in ARL patients not taking ART compared with HIV-negative patients [33]. Barta et al demonstrated that survival increased from 24% pre-ART to 67% in the ART era [22]. All 3 studies show a clear survival benefit to ART, which is not evident in our cohort. A similarly matched South African cohort to our study cohort also showed poor outcome pre-ART [17]. This lack of prognostic significance in our study is therefore most likely due to the smaller number of patients not on ART (n = 16), rather than a true lack of significance.

Although a low or undetectable viral load in our study significantly affected CR (p = 0.01), this did not translate to a significantly improved FPS or OS. Our findings are similar to a Spanish study in terms of viral load correlating with CR but did not match the correlation with virological response and improved OS seen in this Spanish cohort [32]. The lack of impact on the OS in our cohort may be due to small sample size, or lymphocyte dysregulation, even whilst on ART being driven by immune activation [34,35] rather than HIV viral replication [36,37]. This could negate the benefit of a low or undetectable viral load on OS.

The CD4 count was not associated with improved OS (p = 0.787), and this shift away from the significance of the immune status reflected by the CD4 count in the ART era has been well documented [13,22,38]. A retrospective analysis of ARL patients pre and post-ART, found that a low CD4 count of <100cells/mL was associated with poorer survival only in the pre-ART era [21]. However, the prognostic value of the CD4 cell count in the post-ART era is not entirely lost. When used in a composite score with the viral load and past history of AIDS-defining illness, it showed a strong association with mortality [23].

There was no association between the timing of ART and survival in our cohort (aHR 0.85 [95% CI 0.34, 2.11] p = 0.719), which is borne out by a Chinese study on 100 patients with ARL [39]. However, this is contradictory to the multicentre study in sub-Saharan Africa by Gopal et al [40] and a Brazilian study [41] which found a significantly lower OS in ARL diagnosed in patients already on ART. A German study also found poorer OS in ARL patients already on ART at lymphoma diagnosis compared with ART initiated post lymphoma diagnosis [42]. The conflicting results on the prognostic significance of the timing of ART in ARL is worthy of further study.

In this study, there was no improvement in outcome with the use of rituximab, in contrast to the clearly proven benefit in multiple international studies [16,42–44]. Unsurprisingly, patients who received 6–8 cycles of chemotherapy had a significantly better outcome than patients receiving < 6 cycles of chemotherapy (p = <0.001). The FLYER trial demonstrated non-inferiority of 4 cycles of R-CHOP and 2 additional rituximab doses compared to 6 cycles of R-CHOP [45]. However, this group included only HIV-negative patients with DLBCL and an excellent aaIPI of 0. In our cohort, only 1 patient had an aaIPI of 0. This patient had progressive disease on R-CHOP and required radiotherapy and salvage chemotherapy to attain a 4-year OS. Compatible with our study findings, another cohort in Africa found that patients with ARL who received < 3 cycles of chemotherapy had a significant increase in mortality at 18 months [28]. In the absence of randomised trials, and noting that most patients with ARL have an aaIPI >0, our findings would support current guidelines of 6–8 cycles of R-chemotherapy in CD20 positive ARL [46,47].

Although the numbers are small, with only 9% receiving radiotherapy, the trend toward significance in survival is different to another small study on consolidative radiotherapy in HIV DLBCL, which found no significant difference in OS in patients who received radiotherapy [48]. Unlike this latter study, the indications for radiotherapy in our study were more diverse and included bulky disease, relapse and progressive disease. For the only 2 patients who received radiotherapy but did not reach 4-year OS, both had high-risk IPI of 4, with one

patient having additional risk with testicular involvement. There is limited literature on the value of radiotherapy in ARL apart from primary central nervous system lymphoma. Radiotherapy as a therapeutic tool in ARL management merits further investigation.

Both the IPI and aaIPI were associated with OS. The IPI also correlated with PFS (**Fig 2**). These findings are supported by other studies [49–51]. However, unlike the latter studies, the individual components in the aaIPI in our cohort did not correlate with outcome, either in univariate or multivariate analysis. The skewed LDH results, with 96% of the study cohort having a raised LDH, would also account for it not being of predictive value.

## Study limitations

The retrospective design of the study limited the nature and extent of patient and laboratory information available (viral load was only available for 65 patients). Data related to ART regimens and prior diagnosis of AIDS or opportunistic infection were not available for all patients and the significance of these variables could not be determined. The sample size, in assessing the value of rituximab and ART, was a further limitation, that will be addressed in a local, prospective study. The large number of patients lost to follow-up (38%) was a further limitation.

## Conclusion

This study demonstrates reasonable 4-year survival outcomes with combination chemotherapy and ART in CD20 positive ARL. It validates the utility of the IPI and aaIPI in determining prognosis in ARL in an HIV endemic province in South Africa. Further studies are required to explore the prognostic significance of ART timing, gender and the feasibility of individualized chemotherapy as stratified by prognostic factors.

## Supporting information

**S1 Data.**
(ZIP)

## Acknowledgments

Mrs Pam Pillay and Dr Felicia Manickam for data collection and electronic data capture.

The nurses and doctors at the haematology clinic and chemotherapy room at Edward Vlll Hospital.

National Health Laboratory Service and management at King Edward Vlll hospital for integrating laboratory and clinical haematology care in the province.

## Author Contributions

**Conceptualization:** Nadine Rapiti, Mahomed-Yunus S. Moosa.

**Data curation:** Nadine Rapiti, Nada Abdelatif.

**Formal analysis:** Nada Abdelatif, Mahomed-Yunus S. Moosa.

**Funding acquisition:** Nadine Rapiti.

**Investigation:** Nadine Rapiti.

**Methodology:** Nadine Rapiti, Nada Abdelatif, Mahomed-Yunus S. Moosa.

**Project administration:** Nadine Rapiti.

**Resources:** Nadine Rapiti.

**Software:** Nada Abdelatif.

**Supervision:** Mahomed-Yunus S. Moosa.

**Validation:** Nadine Rapiti, Nada Abdelatif, Mahomed-Yunus S. Moosa.

**Visualization:** Nadine Rapiti, Mahomed-Yunus S. Moosa.

**Writing – original draft:** Nadine Rapiti.

**Writing – review & editing:** Nadine Rapiti, Nada Abdelatif, Mahomed-Yunus S. Moosa.

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
