## [Decision Letter · Decision Letter 0]

15 Jun 2022

PONE-D-22-12668Prognostic variables and 4-year survival outcomes in CD20 Positive AIDS Related Lymphoma in the Anti-retroviral treatment era: A Retrospective Review from a Single Centre in KwaZulu-Natal, South AfricaPLOS ONE

Dear Dr Rapiti

Thank you for submitting your manuscript to PLOS ONE. After careful consideration, we feel that it has merit but does not fully meet PLOS ONE’s publication criteria as it currently stands. Therefore, we invite you to submit a revised version of the manuscript that addresses the points raised during the review process.

We look forward to receiving your revised manuscript.

Kind regards,

Zivanai Cuthbert Chapanduka, MBChB (M.D)

Academic Editor

PLOS ONE

Journal Requirements:

Reviewers' comments:

Reviewer's Responses to Questions

**Comments to the Author**

1. Is the manuscript technically sound, and do the data support the conclusions?

Reviewer #1: Partly

Reviewer #2: Yes

2. Has the statistical analysis been performed appropriately and rigorously? 

Reviewer #1: Yes

Reviewer #2: I Don't Know

3. Have the authors made all data underlying the findings in their manuscript fully available?

Reviewer #1: No

Reviewer #2: Yes

4. Is the manuscript presented in an intelligible fashion and written in standard English?

Reviewer #1: Yes

Reviewer #2: Yes

5. Review Comments to the Author

Reviewer #1: Thank you for the opportunity to review thi manuscript entitled "Prognostic variables and 4-year survival outcomes in CD20 Positive AIDS Related Lymphoma in the Anti-retroviral treatment era: A Retrospective Review from a Single Centre in KwaZulu-66 Natal, South Africa".

This research is a well-designed retrospective study highlighting that established scoring systems for aggressive lymphomas are relevant in the context of AIDS related lymphomas. Additional findings were that a low viral load was associated with increased probability of CR and that male gender and 6 or more cycles of chemotherapy were associated with longer survival.

Assessing predictors of lymphoma outcome within the context of an HIV endemic setting is important and will be of interest to the readership of PLOSOne.

It is recommended that the manuscript be accepted for publication with revisions.

The following is unclear and should be addressed:

1. Within the abstract, it states that “attaining a CR and male gender were significantly associate with improved 4-year OS and PFS”. The p-value for male gender is stated. It is not clear where the impact of attaining CR on 4-year OS and PFS is stated; a HR and p-value were not calculated for this variable within Table 3.

2. In Figure 1

a. On the left (CR), there are 56 patients that achieved CR1 with 11 of these lost to follow up after CR1 and 21 patients required salvage therapy (2 of these were patients that had achieved CR1 – from the text), leaving 27 of the 102 patients unaccounted for. The authors should add into the flow chart what happened to these 27 patients. Were they lost to follow up before CR1, did they demise before achieving CR1 or before salvage therapy could be offered, or refuse salvage?

b. On the right (PFS), it should be added that 12 demised and 30 were lost to follow up, as per OS. Or the figure restructured to indicate that the demised/LTFU box is applicable to both OS and PFS.

3. Please elaborate within the method section as to why the cut-off values were chosen for analysis. Why was age <40 years chosen, if the IPI decision point is age <60? All of the other cut-offs are the same as the IPI. Why was CD4 count cut-off of 100 chosen for comparison?

4. In Table 1, the percentage of patients with CD4 count <100 and >100 are not stated, while percentages are stated for other categories. This should be remedied.

5. All data underlying the findings should be made available within the manuscript. The authors should consider adding percentages for each category in Table 2 to Table 4, as done for Table 1, as this information is not stated within the text. Additionally, data on which the interpretation were based may be incomplete. In all tables, it will be useful to add how many patients had available data for each variable. For example, we learn in the limitations that for VL only 65 patients had data. It will be valuable to know an n for all variables analysed, as this has impact on the calculated hazard ratios and p-values.

6. The manuscript should be carefully checked to make sure that all abbreviations are written out on first use.

Reviewer #2: -I suggest a clear definition of CD20 positive AIDS-related lymphomas and why that terminology has been preferred to (CD20 positive) lymphomas associated with HIV infection (as per WHO classification of lymphomas). This is more so given the stated limitation that the diagnosis of AIDS was not confirmed in a sizeable population in this cohort of patients

- In this study, what subtypes of CD20 ARLs were encountered and in which proportions?

- I see that the only chemo regimens used were CHOP or R-CHOP. Were these given at standard doses and frequency or there were dose & frequency modifications? (If dose &/or frequency modifications were done, did that have any impact on PFS or OS?) And were these the regimens also used to treat Burkitt lymphoma and primary CNS lymphoma, if you had any in this patient cohort? And were the other known prognostic factors in Burkitt lymphoma such as CNS involvement or tumour diameter assessed?

- Prognostic value of age >35 (rather than >40 years) is known to be poorer than <35 years for this patient population. What was the literature basis of using 40 years in this study rather than 35 years?

- I've concerns about two important findings in this study that contradict literature in which larger numbers of patients were assessed:

1) Impact of adding rituximab to CHOP - no beneficial effect in this cohort

2) Impact of starting chemo in ART naive vs patients already on ART - no effect of OS

Do you think this is the message you would want the paper to convey to the audience or you think this is artefactual due to small sample size? If the latter is true, don't you think it is better to get a more representative sample size before publishing this article?

- Table 1: please put percentages for patients with CD4 counts below and above 100

- I would think it is important to put the total number of patients with ARL seen during the study period and then state how many were selected for the study, the selection criteria and why the rest were excluded

- I notice that poor virological control had an impact on achieving CR. Why was that so, and what was the impact of confounders such as higher rates of neutropenia, delay in giving chemo as per protocol, compliance to chemo, opportunistic infections, etc? Were these confounding factors similar in those with HIV VL <50 and those with higher viral load and if not, did you test their prognostic impact independently?

- Line 283: You mention IMPROVED 4-year survival: do you have pre-HAART era survival rates in similar patient cohort to compare with? This is more so important given that you didn't find survival benefit in those patients on ART versus those not on ART when given chemo

- Line 349 may need to be relooked

- I suggest that you include absolute numbers in your results section for the readers to get context of some of your discussion points. e.g., you seem to be recommending readers to utilise radiotherapy more, but it is not clear how many of your patients received it and for what indication or stage of disease, and at palliative or curative doses.. If the numbers are small, your seemingly positive vibe with radiotherapy may just have been by chance.

6. PLOS authors have the option to publish the peer review history of their article (what does this mean?). If published, this will include your full peer review and any attached files.

Reviewer #1: No

Reviewer #2: No

---

## [Author Response · Author response to Decision Letter 0]

27 Jun 2022

Response to Reviewers PlosOne 27 June 2022

Reviewer #1: Thank you for the opportunity to review this manuscript entitled "Prognostic variables and 4-year survival outcomes in CD20 Positive AIDS Related Lymphoma in the Anti-retroviral treatment era: A Retrospective Review from a Single Centre in KwaZulu-66 Natal, South Africa".

This research is a well-designed retrospective study highlighting that established scoring systems for aggressive lymphomas are relevant in the context of AIDS related lymphomas. Additional findings were that a low viral load was associated with increased probability of CR and that male gender and 6 or more cycles of chemotherapy were associated with longer survival.

Assessing predictors of lymphoma outcome within the context of an HIV endemic setting is important and will be of interest to the readership of PLOSOne.

It is recommended that the manuscript be accepted for publication with revisions.

The following is unclear and should be addressed:

1. Within the abstract, it states that “attaining a CR and male gender were significantly associate with improved 4-year OS and PFS”. The p-value for male gender is stated. It is not clear where the impact of attaining CR on 4-year OS and PFS is stated; a HR and p-value were not calculated for this variable within Table 3.

Response: Thank you for pointing out this omission; it has now been corrected. “Attaining a CR and male gender were significantly associated with improved 4-year OS (p<0.001 and p=0.028 respectively) and PFS (p<0.001 and 0.048 respectively).” has been amended in the abstract and “Attaining a CR was significantly associated with improved PFS (HR = 0.08 �CI: 0.04, 0.16�; p-value <0.001) and OS (HR = 0.11 �CI: 0.06, 0.22)�; p-value <0.001).” has been added to the Results section. 

2. In Figure 1

a. On the left (CR), there are 56 patients that achieved CR1 with 11 of these lost to follow up after CR1 and 21 patients required salvage therapy (2 of these were patients that had achieved CR1 – from the text), leaving 27 of the 102 patients unaccounted for. The authors should add into the flow chart what happened to these 27 patients. Were they lost to follow up before CR1, did they demise before achieving CR1 or before salvage therapy could be offered, or refuse salvage?

Response: At at end of 1st line chemotherapy, 56 patients were in CR, 7 had a partial response, 19 had disease progression, 9 demised and 11 were lost to follow up. This has now been included in Fig 1.

b. On the right (PFS), it should be added that 12 demised and 30 were lost to follow up, as per OS. Or the figure restructured to indicate that the demised/LTFU box is applicable to both OS and PFS.

Response: A box has been added to include these details ie. 30 were LTFU, 12 demised and 7 had disease progression. 

3. Please elaborate within the method section as to why the cut-off values were chosen for analysis. Why was age <40 years chosen, if the IPI decision point is age <60? All of the other cut-offs are the same as the IPI. Why was CD4 count cut-off of 100 chosen for comparison?

Response: There was only 1 patient in the cohort who was 60 years old. All other patients were <60years. Manyau et al (Zimbabwe)32 assessing the impact of rituximab in ARL, found poorer outcome with age � 40 years.

In another South African paper by de Witt et al25, a CD4 of 100 was chosen as a comparison. A German study by Wyen et al41 found better PFS and OS with CD4 counts ≥100 cells/µL . In the HIV score by Mounier et al which was a third of the Barta et al16 cohort), a point was assigned for CD4 count < 100 cells/µL. In the Chinese analysis by Lim et al21, a CD4 count of <100 predicted poorer OS in the pre-ART era.

This clarity has now been added in Methods. “The cut-off for the age of 40 and CD4 of 100 cells/µL was chosen to allow for comparison with existing data, which shows prognostic significance to these variables at these limits.16,21,25,32”.

4. In Table 1, the percentage of patients with CD4 count <100 and >100 are not stated, while percentages are stated for other categories. This should be remedied.

Response: This omission has now been corrected in Table 1 CD4<100=26=25%, ≥ 100 =76 (75%)

5. All data underlying the findings should be made available within the manuscript. The authors should consider adding percentages for each category in Table 2 to Table 4, as done for Table 1, as this information is not stated within the text. Additionally, data on which the interpretation were based may be incomplete. In all tables, it will be useful to add how many patients had available data for each variable. For example, we learn in the limitations that for VL only 65 patients had data. It will be valuable to know an n for all variables analysed, as this has impact on the calculated hazard ratios and p-values.

Response: All data was available for all patients, apart from the viral load, hence it was stated as a limitation. This has also now been included in the Methods section “All records were available, except for viral load, where only 65 patients had information.” 

The percentages have been included in Tables 2-4.

6. The manuscript should be carefully checked to make sure that all abbreviations are written out on first use.

Response: The abbreviations have been checked and corrected.

Reviewer #2: -

1.I suggest a clear definition of CD20 positive AIDS-related lymphomas and why that terminology has been preferred to (CD20 positive) lymphomas associated with HIV infection (as per WHO classification of lymphomas). This is more so given the stated limitation that the diagnosis of AIDS was not confirmed in a sizeable population in this cohort of patients

- In this study, what subtypes of CD20 ARLs were encountered and in which proportions?

Response: The reviewer’s interpretation is correct. A pre-existing diagnosis of AIDS (prior to lymphoma diagnosis) could not be established for most patients. However, all lymphomas included in this cohort were AIDS-defining, so AIDS was confirmed on diagnosis of the lymphoma. A better definition has now been included in the introduction “Lymphomas that occur more frequently in people living with HIV, are referred to as HIV-associated lymphomas (HAL). Three malignancies are considered AIDS-defining: high-grade non-Hodgkin lymphoma (NHL), Kaposi sarcoma and invasive cancer of the cervix. The AIDS-defining high-grade NHL are collectively called AIDS-related lymphomas (ARL) and are aggressive with diverse histologic characteristics.1 These ARL include diffuse large B cell lymphoma (DLBCL), Burkitt lymphoma (BL), plasmablastic lymphoma (PBL) and less commonly, primary CNS lymphoma (PCNSL) and primary effusion lymphoma (PEL).2,3 DLBCL and BL usually express CD20, and the other ARL are generally CD20 negative.”

The histological detail has now been included in the Results. “There were 102 patients with CD20 positive ARL included in this analysis (histological subtypes were 70 DLBCL, 31 high-grade B cell lymphoma, and 1 high-grade B cell lymphoma not otherwise specified)” 

- I see that the only chemo regimens used were CHOP or R-CHOP. Were these given at standard doses and frequency or there were dose & frequency modifications? (If dose &/or frequency modifications were done, did that have any impact on PFS or OS?) And were these the regimens also used to treat Burkitt lymphoma and primary CNS lymphoma, if you had any in this patient cohort? And were the other known prognostic factors in Burkitt lymphoma such as CNS involvement or tumour diameter assessed?

Response: The cohort did not include any patients with Burkitt lymphoma, as this lymphoma in not managed in this hospital(more intensive chemotherapy at another institution). PCNSL was not included in this description. This is now added in the Methods section: “Patients with Burkitt lymphoma were not included in this study, as these patients are treated with a more intensive chemotherapy protocol. Patients with primary CNS lymphoma were excluded from this analysis”

The only 2 regimens used were CHOP and R-CHOP, in standard doses, apart from Prednisone which is given at 60mg x 8 days, rather than 100mg x 5 days. This has now been added in the Methods section “Patients were treated with standard doses of CHOP and RCHOP, apart from prednisone, which was given at a dose of 60mg daily orally for 8 days, instead of 100mg for 5 days, according to the centre practice.”

- Prognostic value of age >35 (rather than >40 years) is known to be poorer than <35 years for this patient population. What was the literature basis of using 40 years in this study rather than 35 years?

Response: This was to allow comparison with another study from Africa, by Manyau et al (Zimbabwe)32 who in assessing the impact of rituximab in ARL, found poorer outcome with age � 40 years. This has now been clarified in the Methods section “The cut-off for the age of 40 and CD4 of 100 cells/µL was chosen to allow for comparison with existing data, which shows prognostic significance to these variables at these limits.16,21,25,32”.

- I've concerns about two important findings in this study that contradict literature in which larger numbers of patients were assessed:

1) Impact of adding rituximab to CHOP - no beneficial effect in this cohort

Response: This concern is noted. This will be addressed more clearly in the Discussion: “In this study, there was no improvement in outcome with the use of rituximab, in contrast to the clearly proven benefit in multiple international studies.16,41,42,43 and in the Study limitations as follows: “The sample size, in assessing the value of rituximab and ART, was a further limitation, that will be addressed in a local prospective study.”

2) Impact of starting chemo in ART naive vs patients already on ART - no effect of OS

Do you think this is the message you would want the paper to convey to the audience or you think this is artefactual due to small sample size? If the latter is true, don't you think it is better to get a more representative sample size before publishing this article?

Response: This lack of prognostic significance will be re-worded more strongly in our discussion, as follows: “This lack of prognostic significance in our study may be due to the smaller number of patients not on ART (n=16).” has now been changed to: “Barta et al demonstrated that survival increased from 24% pre-ART to 67% in the ART era.22 All 3 studies show a clear survival benefit to ART, which is not evident in our cohort. A similarly matched South African cohort to our study cohort also showed poor outcome pre-ART.17 This lack of prognostic significance in our study is therefore most likely due to the smaller number of patients not on ART (n=16), rather than a true lack of significance.” 

This will also be added as a study limitation “The sample size, in assessing the value of rituximab and ART, was a further limitation, that will be addressed in a local prospective study.”

- Table 1: please put percentages for patients with CD4 counts below and above 100

Response: This has now been included.

- I would think it is important to put the total number of patients with ARL seen during the study period and then state how many were selected for the study, the selection criteria and why the rest were excluded

Response: Now included in Methods: “Patients with Burkitt lymphoma were not included in this study, as these patients are treated with a more intensive chemotherapy protocol. Patients with primary CNS lymphoma were excluded from this analysis” and in Results “During the study time period, 157 ARL were managed at this hospital. There were 102 patients with CD20 positive ARL included in this analysis (histological subtypes were 70 DLBCL, 31 high-grade B cell lymphoma, and 1 high-grade B cell lymphoma not otherwise specified). 13 patients were excluded due to missing data (either histology or HIV results), 9 patients for receiving an alternative chemotherapy regimen, 5 patients who did not receive chemotherapy and 2 patients with primary effusion lymphoma. The data for 26 plasmablastic patients has been published”

- I notice that poor virological control had an impact on achieving CR. Why was that so, and what was the impact of confounders such as higher rates of neutropenia, delay in giving chemo as per protocol, compliance to chemo, opportunistic infections, etc? Were these confounding factors similar in those with HIV VL <50 and those with higher viral load and if not, did you test their prognostic impact independently?

Response: The neutropenic rate and infections in patients with poor virological control were not compared with those with good virological control. There was no difference or delay with chemotherapy in the protocol according to virological response. These confounding factors will be studied further prospectively.

- Line 283: You mention IMPROVED 4-year survival: do you have pre-HAART era survival rates in similar patient cohort to compare with? This is more so important given that you didn't find survival benefit in those patients on ART versus those not on ART when given chemo. 

Response: A South African study by Patel et al 17 showed a median OS of 11 months in a cohort of 198 ARL, with 85% (169/198)of patients not on ART. In the De Witt25 paper (also from South Africa), 78% of the patients were on ART and the 2 year OS was 40.5%. Of the prognostic variables assessed, good response to ART predicted a statistically significant 2 year OS and PFS ie. (54% versus 18% 2-year OS, p=0.03) and PFS 45% vs 14% (p=0.03). 

This has now been revised in the manuscript Discussion: “This improved 4-year survival in the ART era, compared with the 11-month median OS described in Gauteng, South Africa with 85% of the cohort not on ART17, highlights the need for an accurate prognostic system to guide therapy in ARL”

- Line 349 may need to be relooked

- I suggest that you include absolute numbers in your results section for the readers to get context of some of your discussion points. e.g., you seem to be recommending readers to utilise radiotherapy more, but it is not clear how many of your patients received it and for what indication or stage of disease, and at palliative or curative doses.. If the numbers are small, your seemingly positive vibe with radiotherapy may just have been by chance.

Response: This has now been moved to the Results section, and edited to include numbers and indications. “Of the total cohort, 9% (n=9) received radiotherapy; 4 patients with bulky disease, 1 patient with a tonsillar relapse (who received salvage chemotherapy and radiotherapy) and progressive disease treated with salvage chemotherapy in the remaining four patients. A third of our study patients who received radiotherapy (3/9) had a good response with the sole addition of curative dose radiotherapy to 1st line chemotherapy, converting partial responses to CR in all three patients (all 3 had bulky disease). There was a 78% (7/9) 4-year OS for patients receiving radiotherapy compared with 47% (44/93) for patients who did not receive radiotherapy (p=0.071).”

The line in the Discussion has been re-worded: “Although the numbers are small, with only 9% receiving radiotherapy, the trend toward significance in survival is …” and re-worded more cautiously “Radiotherapy as a therapeutic tool in ARL management merits further investigation.”

---

## [Decision Letter · Decision Letter 1]

18 Jul 2022

Prognostic variables and 4-year survival outcomes in CD20 Positive AIDS Related Lymphoma in the Anti-retroviral treatment era: A Retrospective Review from a Single Centre in KwaZulu-Natal, South Africa

PONE-D-22-12668R1

Dear Dr. Rapiti

We are pleased to inform you that your manuscript has been judged scientifically suitable for publication and will be formally accepted for publication once it meets all outstanding technical requirements.

Kind regards,

Zivanai Cuthbert Chapanduka, MBChB (M.D)

Academic Editor

PLOS ONE

Additional Editor Comments (optional):

Congratulations on the acceptance of your manuscript by both reviewers who expressed satisfaction with your responses. If you find anything that must be corrected, please point it out to the Academic Editor or other Editor as soon as possible. 

Reviewers' comments:

Reviewer's Responses to Questions

**Comments to the Author**

Reviewer #1: All comments have been addressed

Reviewer #2: All comments have been addressed

2. Is the manuscript technically sound, and do the data support the conclusions?

Reviewer #1: Yes

Reviewer #2: (No Response)

Academic editor opinion: Yes

3. Has the statistical analysis been performed appropriately and rigorously? 

Reviewer #1: Yes

Reviewer #2: (No Response)

Academic editor opinion: Yes

4. Have the authors made all data underlying the findings in their manuscript fully available?

Reviewer #1: Yes

Reviewer #2: (No Response)

5. Is the manuscript presented in an intelligible fashion and written in standard English?

Reviewer #1: Yes

Reviewer #2: (No Response)

Academic editor opinion: Yes

6. Review Comments to the Author

Reviewer #1: The authors have addressed and clarified all previous concerns.

Line 211: "The data for 26 plasmablastic lymphoma patients has been published." Please supply the reference.

Reviewer #2: (No Response)

7. PLOS authors have the option to publish the peer review history of their article (what does this mean?). If published, this will include your full peer review and any attached files.

Reviewer #1: No

Reviewer #2: No

---

## [Editor Report · Acceptance letter]

23 Aug 2022

PONE-D-22-12668R1 

Prognostic variables and 4-year survival outcomes in CD20 Positive AIDS-Related Lymphoma in the Anti-retroviral treatment era: A Retrospective Review from a Single Centre in KwaZulu-Natal, South Africa 

Dear Dr. Rapiti:

I'm pleased to inform you that your manuscript has been deemed suitable for publication in PLOS ONE. Congratulations! Your manuscript is now with our production department. 

Kind regards, 

on behalf of

Dr. Zivanai Cuthbert Chapanduka 

Academic Editor

PLOS ONE